# Cryptic Diversity in *Cladosporium cladosporioides* Resulting from Sequence-Based Species Delimitation Analyses

**DOI:** 10.3390/pathogens10091167

**Published:** 2021-09-10

**Authors:** Andrea Becchimanzi, Beata Zimowska, Rosario Nicoletti

**Affiliations:** 1Department of Agricultural Sciences, University of Naples Federico II, 80055 Portici, Italy; andrea.becchimanzi@unina.it (A.B.); rosario.nicoletti@crea.gov.it (R.N.); 2Department of Plant Protection, University of Life Sciences, 20-069 Lublin, Poland; 3Council for Agricultural Research and Economics, Research Centre for Olive, Fruit and Citrus Crops, 81100 Caserta, Italy

**Keywords:** *Cladosporium*, cryptic species, phylogenetic analysis, species delimitation methods, taxonomic markers

## Abstract

*Cladosporium cladosporioides* is an extremely widespread fungus involved in associations ranging from mutualistic to pathogenic and is the most frequently represented *Cladosporium* species in sequence databases, such as Genbank. The taxonomy of *Cladosporium* species, currently based on the integration of molecular data with morphological and cultural characters, is in frequent need of revision. Hence, the recently developed species delimitation methods can be helpful to explore cryptic diversity in this genus. Considering a previous study that reported several hypothetical species within *C. cladosporioides*, we tested four methods of species delimitation using the combined DNA barcodes internal transcribed spacers, translation elongation factor 1-α and actin 1. The analyses involved 105 isolates, revealing that currently available sequences of *C. cladosporioides* in GenBank actually represent more than one species. Moreover, we found that eight isolates from this set should be ascribed to *Cladosporium anthropophilum*. Our results revealed a certain degree of discordance among species delimitation methods, which can be efficiently treated using conservative approaches in order to minimize the risk of considering false positives.

## 1. Introduction

Fungi belonging to the genus *Cladosporium* (Dothideomycetes, Cladosporiaceae) are ubiquitous in connection with their ability to colonize any kind of organic substrate in both terrestrial and marine environments [1]. With their branched chains of small conidia, which are easily spread over long distances, *Cladosporium* species represent the most common fungi isolated from the air [2]. Other species are pathogenic to plants and animals, hyperparasites of other fungi, or common epiphytes and endophytes [2,3,4]. Their environmental plasticity and capacity to establish successful biocenotic interactions are also supported by peculiar biosynthetic capacities, which also make these fungi an interesting source of novel bioactive compounds [5].

The taxonomy of *Cladosporium* is constantly evolving after recent revisions have pointed out that morphological characters need to be integrated with molecular and ecological data in the attempt to go further into the typification of the many cryptic species which have recently been identified [1,6,7,8,9]. More than 230 species are currently recognized in this genus, which is subdivided into three main species complexes: *C. cladosporioides*, *C. herbarum* and *C. sphaerospermum* [6]. The assignment of *Cladosporium* isolates to one of these major species complexes is usually based on morphology or internal transcribed spacers (ITS) sequence analysis [4]. However, phylogenetic reconstructions carried out by using other molecular markers, such as translation elongation factor 1-α (*tef1*) and actin 1 (*act*), have revealed the species limits of *Cladosporium* and improved the understanding of the hidden diversity within this genus [3,10].

In *Cladosporium,* limited insights have been done through DNA-based species delimitation methods, although these tools are recommended as part of an integrative approach to establish well-supported boundaries among fungal species [9,11]. These methods use distinct strategies, including genetic distance and coalescence [9], and are increasingly employed in fungal taxonomy [12,13,14].

We recently examined a set of *Cladosporium* isolates associated with galled and non-galled flowers of several plants belonging to the Lamiaceae and found it to consist in an assortment of at least 10 species within the *C. cladosporioides* and *C. herbarum* species complexes [15]. Two novel species belonging to the first taxonomic group were described, and the existence of a wide genetic variation was observed among the isolates ascribed to the species *C. pseudocladosporioides* and *C. cladosporioides*. In particular, the use of two species delimitation methods indicated the possible existence of additional species to be identified within the latter. Defined as the founder of the homonymous species complex, *C. cladosporioides* is extremely widespread in both terrestrial and marine environments where it is found as a symbiont of many plants and animals in associations ranging from mutualistic to pathogenic [1,16]. It also represents the most common *Cladosporium* species according to the literature and the number of strains having their DNA sequences deposited in GenBank. Based on the provisional evidence resulting in our previous study [15], we decided to more accurately investigate the phylogenetic relationships among the strains of *C. cladosporioides* that have so far been genetically characterized through the deposit in GenBank of sequences of the taxonomic markers ITS, *tef1* and *act*, which are required for molecular delimitation at species level by multilocus approach [2].

We performed phylogenetic and species delimitation analyses aimed at exploring the cryptic diversity of *C. cladosporioides*. In the course of these analyses, we also assessed intron presence/absence in *tef1* sequences, which is considered a feature of phylogenetic importance and often reported as lineage-specific [17]. We analyzed our dataset through different species delimitation methods and placed our trust in delimitations that are congruent across methods. One sequence-based and three tree-based methods were employed that are among the most popular approaches for species delimitation based on sequence data and are frequently used in studies on fungal diversity [12,13,14,18]: the automatic barcode gap discovery (ABGD) [19], the general mixed Yule-coalescent (GMYC) model [20,21], the Poisson Tree Processes (PTP) [22] and its multi-rate extension (mPTP) [23].

ABGD is a sequence-based method that sorts the sequences into hypothetical species based on the barcode gap, which can be observed whenever intra-specific is smaller than inter-specific divergence [19]. This is a fast method to split a sequence alignment dataset into candidate species, but its output should be interpreted by complementation with other methods [19]. The GMYC model uses maximum likelihood and an ultrametric gene tree to model the transition between inter- and intraspecific branching patterns [21]. Indeed, this method is based on the prediction that independent evolution leads to the appearance of distinct genetic clusters, separated by longer internal branches in a gene tree [20]. Likewise, PTP tries to determine the transition point from a between- to a within-species process using a two-parameter model, one for the speciation and one for the coalescent process [22]. In contrast to GMYC, PTP estimates branching processes using the expected number of substitutions (vs. time in GMYC) and thus exploits a non-ultrametric phylogenetic tree as input [23]. PTP assumes that every species evolved with the same rate in phylogeny; however, this generally disregards the stochastic variation among species due to different population sizes and demographic histories. Conversely, the recently developed mPTP fits the branching events of each delimited species to a distinct exponential distribution to account for differences in sampling intensity [23].

To our knowledge, this is the first study that evaluates the performance of several species delimitation methods in *Cladosporium* and provides a useful framework for combining different analyses aimed at identifying cryptic diversity in fungi.

## 2. Results

### 2.1. Phylogenetic Analysis

Overall, our analysis included 127 strains (105 of which are reported as *C. cladosporioides*) and was based on a nucleotide set of ~1400 bp (~690 bp for ITS, ~490 bp for *tef1* and ~220 bp for *act*) (Table 1 and Table 2). The resulting alignment revealed the presence of a ~60 bp intron in the *tef1* sequence. These isolates formed seven groups with high ML/MP bootstrap support values (82–100), with the majority of them clustering into groups from A to E (Figure 1).

Group A is the largest group, made of 37 isolates, including the neotype CBS 112388 and other strains usually employed as references for this species in phylogenetic studies, besides miscellaneous isolates from diverse locations (Brazil, Europe, Australia, USA and China) and sources (indoor environment, plants and human tissues). Groups B, C, D and E are closely related to group A but do not include any reference strain. Group B is formed by 14 isolates with different origins (algae from Portugal, rice leaf from Brazil, fruits from South Africa, etc.). Notably, only four of these haplotypes present introns in the *tef1* sequence. In group C, there are seven isolates from plants of Korea and India. Isolates in group D have been collected from different geographical and ecological sources (indoor air from the USA, wheat from South Africa, fruits from Mexico, cecidomyid galls from Italy, etc.). Interestingly, 16 out of 26 isolates in this group contain introns in the *tef1* sequence (Figure 1). Group E is clearly divergent from the previous groups and only includes two isolates: AjNa1 from the flower of *Ajuga reptans* in Italy, which was obtained in our previous work [15], and CPC 15626 from an unspecified ‘wild’ plant in Mexico. Seven out of eight isolates in group F were recovered from conipherous plants in Korea, while one (CBS 674.82) was obtained from cotton seeds in Israel. These isolates cluster together with representatives of the recently described *C. anthropophilum* [22]. Finally, group G is formed of two isolates from leaves of *Camellia sinensis* collected in southwest China and six isolates from Korea associated with several plants and with the Japanese pear rust (*Gymnosporangium asiaticum,* current name of *G. haraeanum*). Notably, the phylogram shows that the latter two groups are less closely related to *C. cladosporioides* than to the other species included in the analysis, namely *C. tenuissimum, C. colocasiae, C. oxysporum, C. vignae, C. angustisporum, C. subuliforme* and *C. cucumerinum*.

### 2.2. Species Delimitation Analyses

The same aligned dataset was analyzed using four different species delimitation methods. mPTP was the most conservative method, inferring only four species within *C. cladosporioides* (groups A to E, F and G; Figure 2). ABGD was the second most conservative method, identifying five species, while GMYC and PTP respectively detected 11 and 12 species within *C. cladosporioides*.

The tested species delimitation methods correctly identified known species, except for mPTP, which failed to discriminate among *C. tenuissimum, C. colocasiae, C. oxysporum, C. vignae, C. angustisporum, C. subuliforme* and *C. cucumerinum*, as well as among *C. xylophilum*, *C. neapolitanum* and *C. rectoides* (Figure 2). The consensus among the remaining methods is high in the lower-middle part of the tree, where group F (*C. anthropophilum*), group G and the isolate CPC 10142 are indicated as different species (Figure 2). Moreover, isolates AjNa1 and CPC 15626 (group E) are indicated as a single species by ABGD and GMYC, two different species by PTP and grouped together with groups A−D by mPTP. In the upper part of the tree, the methods are discordant, indicating 1 to 7 species for groups A to D, which were pointed out by our phylogenetic reconstruction (Figure 2). The results are very similar between ABGD and mPTP, indicating one species, as well as between GMYC and PTP, which indicated seven species, including the groups A−D (Figure 2).

Overall, the congruence of methods was highest between GMYC and PTP (*C*_tax_ = 0.87). The lowest congruence of methods was observed between GMYC and mPTP (*C*_tax_ = 0.27) and between PTP and mPTP (*C*_tax_ = 0.28). Notably, GMYC has the highest mean index of congruence (*C*_tax_ = 0.62), while mPTP has the lowest mean (0.32) (Table 3).

## 3. Discussion

In this study, we explored cryptic diversity among *C. cladosporioides* strains for which ITS, *tef1* and *act* sequences are available in GenBank (last accessed in May 2021), combining phylogenetic and species delimitation analyses. As shown by the phylogenetic tree, the combination of the above-mentioned loci is reliable to distinguish the currently accepted species and indicates with bootstrap support that the isolates recorded in GenBank as *C. cladosporioides* from seven discrete groups.

Apical groups (A to E) are more closely related to reference strains of *C. cladosporioides*, while basal groups (F and G) are less closely related to *C. cladosporioides* than to *C. tenuissimum, C. colocasiae, C. oxysporum, C. vignae, C. angustisporum, C. subuliforme* and *C. cucumerinum*, suggesting compromised taxonomic annotations. Indeed, group F includes two strains of *C. anthropophilum* and is consistently reported as a different species by all tested species delimitation methods, as well as group G. The presence in public repositories of mismatches between gene sequences and the species names assigned to the isolates from which they were obtained is frequently reported [24,25] and represents a source of potentially propagating errors [26]. Moreover, the development of new species delimitation methods and the fast growth of the number of described species require more efforts for maintaining and updating public databases [27]. As an example, sequences of CBS 674.82 were uploaded in GenBank in 2010 after it was identified as *C. cladosporioides* [3], while *C. anthropophilum* was described for the first time only in 2016 [4].

Our phylogenetic reconstruction pointed out that no clear association can be inferred between plant species, or geographic areas, and group compositions. However, we observed a certain correspondence concerning intron presence in *tef1* locus in groups B (28.5% of the isolates) and D (61.5% of the isolates), although the presence/absence of the introns in the partial *tef1* does not follow the geographical distribution of isolates in contrast to a recently published study on the halotolerant fungus *Hortaea werneckii* [28].

The ultrametric tree obtained through the Bayesian approach shows a highly similar topology to the one depicted by the ML tree, confirming the existence of the same seven groups (A−G). Notably, both tree- (GMYC, PTP) and sequence-based (ABGD) methods are able to correctly distinguish the currently accepted species, also in a context of uneven sampling (i.e., isolates per species ranging from 1 to 105). Surprisingly, the multi-rate extension of PTP (mPTP) cannot discriminate among known species revealing an exaggerated lumping tendency in our conditions. Indeed, this method greatly differs from the others in terms of mean values of the *C*_tax_ index, which is a measure of reciprocal congruence between methods, revealing poor delimitation performance for the considered species. These findings are in line with a previous work that suggested that the mPTP method is more conservative than GMYC [25]. Species delimitation analysis unequivocally ascribes groups F and G, as well as the singleton CPC 10142, to different species, with a 100% consensus among methods. In particular, isolates in group F are to be ascribed to *C. anthropophilum*, while isolates in group G clearly represent an unknown species. To assess if isolates of group G could eventually be ascribed to a species that is not included in our tree, their sequences of all the tested loci were blasted against GenBank nr database, obtaining only *C. cladosporiodes* and *C. anthropophilum* as matches. Hence, considering the output of our species delimitation analyses, these isolates cannot be ascribed to any already described species.

Only a partial match of methods is observed for group E (isolates AjNa1 and CPC 15626). mPTP indicates this group as a single species together with groups A to D; conversely, GMYC and ABGD suggest that group E represents a different species, while PTP identified two separate species. This group is placed at the boundary of *C. cladosporioides*, as already observed [15], and represents a potential cryptic taxon worth further studies.

A certain discordance among methods is reported for groups A to D. For these groups, the most conservative methods (ABGD and mPTP) indicate one species, while GMYC and PTP indicate seven species. These results are in line with many studies reporting the tendency of ABGD and mPTP to collapse multiple taxa into one [29,30], as well as the tendency of GMYC and PTP to split [23,31].

How can we manage such discordance? Many authors suggest that a conservative approach is preferable to minimize the risk of oversplitting (i.e., the inclusion of false positives) and, thus, delimiting entities that do not represent actual evolutionary lineages [11,32,33]. Other authors suggest the inclusion of an allopatry/sympatry evaluation in order to assess reproductive barriers in a population [34]. However, applying these criteria to fungi can be problematic (e.g., parasites specialized on different sympatric hosts are sometimes considered allopatric) and requires further investigations [35].

Considering that such a high number of species identified by PTP and GMYC is likely the result of the tendency of these approaches to overestimate the number of species, we adopted a conservative strategy consisting of classifying putative species as the most comprehensive groups of isolates predicted by any of the four delimitation methods. Overall, excluding group F, which is ascribed to *C. anthropophilum*, our species delimitation analysis indicates that, following a conservative approach, *C. cladosporioides* isolates available in GenBank have to be grouped at least in two species.

## 4. Materials and Methods

### 4.1. Phylogenetic Analysis

We selected 105 isolates from GenBank reported as *C. cladosporioides* (Table 1) and 22 isolates ascribed to the 11 most closely related species (*C. neapolitanum, C. rectoides, C. xylophilum, C. tenuissimum, C. colocasiae, C. oxysporum, C. vignae, C. angustisporum, C. anthropophilum, C. subuliforme* and *C. cucumerinum*) and to *C. hillianum*, which was used as the outgroup (Table 2). These species were included in order (1) to provide the taxonomic assignment for *C. cladosporioides* isolates in the GenBank database and (2) to better fit in the context of species delimitation methods (such as the general mixed Yule-coalescent model), which can be destabilized when less than 5 species are included in the analysis [31].

The combined ITS, *tef1* and *act* sequences were aligned by using Muscle [36] and manually adjusted with AliView software version 1.27 [37], where necessary. The aligned sequences were manually checked in order to identify introns, which are frequent in *tef1* [17] and are characterized by the presence of GT-AG nucleotides (5′-3′). Gaps were treated as missing characters. The phylogenetic analyses were carried out in conformity with recent protocols [8,38]. The best nucleotide substitution model (generalized time-reversible model with gamma distribution and a portion of invariable sites (GTR + G + I) for the three independent data sets) was estimated using jModelTest version 2.3 [39] following the Akaike criterion. Phylogenetic analyses of the concatenated sequence data for maximum likelihood (ML) were performed by using RAxML software version 8.2.12 [40] with the GTR + G + I model of nucleotide substitution and 1000 bootstrap replications. Concatenated sequences were also analyzed for maximum parsimony (MP) by using PAUP, under the heuristic search parameters with tree bisection reconnection branch swapping, 100 random sequence additions, maxtrees set up to 1000 and 1000 bootstrap. Bayesian analyses were done with a Markov chain Monte Carlo (MCMC) coalescent approach implemented in BEAST v.2.0.2 [41], using the uncorrelated lognormal relaxed clock, the GTR + G + I model, and a coalescent tree prior. Bayesian MCMC was run for 50 million generations, and trees and parameters were sampled every 1000 generations. The resulting log files were entered in Tracer v1.6.0 to check trace plots for convergence and effective sample size (ESS). Burn-in was adjusted to achieve ESS values of ≥200 for the majority of the sampled parameters. While removing a portion of each run as burn-in, log files and trees files were combined in LogCombiner. TreeAnnotator was used to generate consensus trees with 25% burn-in and to infer the maximum clade credibility tree, with the highest product of individual clade posterior probabilities. Phylogenetic trees were drawn by using FigTree software (tree.bio.ed.ac.uk/software/figtree/, accessed on 12 November 2020).

### 4.2. Species Delimitation Analysis

The ABGD method was tested through a web interface (abgd web, bioinfo.mnhn.fr/abi/public/abgd/abgdweb.html, accessed on 10 May 2021). Before analysis, the model criteria were set as follows: variability (*P*) between 0.001 (*P*min) and 0.1 (*P*max), minimum gap width (×) of 0.1, Kimura-2-parameters and 50 screening steps. To perform the GMYC delimitation method, an ultrametric tree was constructed in BEAST 2, as described above. After removing 25% of the trees as burn-in, the remaining trees were used to generate a single summarized tree in TreeAnnotator v.2.0.2 (part of the BEAST v.2.0.2 package) as an input file for GMYC analyses. The GMYC analyses with a single threshold model were performed in R (R Development Core Team, www.R-project.org, accessed on 12 May 2021) under the “splits” package using the “gmyc” function (R-Forge, r-forge.r-project.org/projects/splits/, accessed on 9 May 2021). The PTP analysis was carried out with the web service available at http://mPTP.h-its.org (accessed on May 2021) under maximum-likelihood estimations, using both PTP (i.e., using the -single ML option) and mPTP (i.e., using the -multi ML option) model. For PTP/mPTP, we used as input the tree produced with RAxML, as described above, and default settings. Finally, we quantified the performance of methods using the Taxonomic Index of Congruence (*C*_tax_) [42]. The *C*_tax_ index is a measure of congruence in species assignments among two methods, with a value of 1 indicating complete congruence. *C*_tax_ metrics were calculated as follows:$$C_{tax}\left(AB\right)=\frac{n\;\left(A\;\cap \;B\right)}{n\;\left(A\;\cup \;B\right)}$$
where *A* ∩ *B* represents the number of speciation events shared by methods *A* and *B*, and *A* ∪ *B* represents the total number of speciation events inferred by method *A* and/or *B* [43].

## 5. Conclusions

Concordance among our results suggests that several strains of *C. cladosporioides* (group G), isolated in Korea and China, represent a new putative species that requires morphological characterization prior to formal taxonomic changes. A certain degree of cryptic diversity was observed for group E (isolates AjNa1 and CPC 15626); however, following our conservative approach, this two-membered group cannot be ascribed to a new species without additional molecular, morphological and ecological characterization. Combining in a single analysis several criteria of species delimitation likely brings out discordance among methods, which can be efficiently treated using conservative approaches in order to minimize the risk of considering false positives. Such a promising strategy represents a precious tool for elucidating diversity in directly collected specimens, as well as in public repositories of molecular data.

Moreover, we reported an erroneous taxonomic annotation in GenBank for isolates in group F, which should be ascribed to *C. anthropophilum* according to the current taxonomic arrangement. Indeed, data recorded in GenBank require careful examination before being used for taxonomic purposes. Nevertheless, public repositories represent a fundamental resource for studying cryptic diversity using molecular data, which can be viewed as a first step for delineating new taxonomic entities in the highly diverse realm of fungi.

## Figures and Tables

**Figure 1 pathogens-10-01167-f001:**
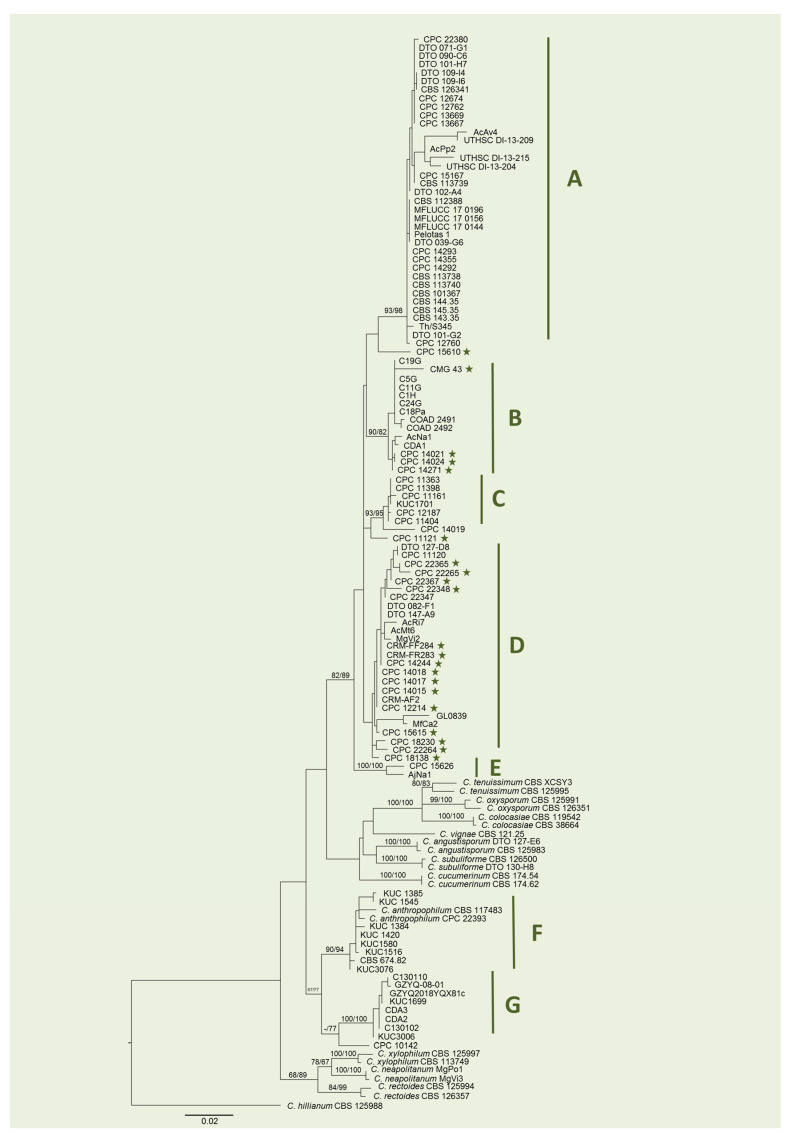
Phylogenetic tree based on maximum likelihood (ML) analysis of combined ITS, *tef1* and *act* sequences of 123 strains from the *C. cladosporioides* complex. Bootstrap support values ≥60% for ML and maximum parsimony (MP) are presented above branches as follows: ML/MP; bootstrap values <60% are marked with ‘-’. *C. hillianum* CBS 12598 was used as an outgroup reference. Highly supported groups are indicated by letters A, B, C, D, E, F and G. Stars indicate the presence of a 60 bp intron in *tef1* sequence. The scale bar indicates the number of nucleotide substitutions per site.

**Figure 2 pathogens-10-01167-f002:**
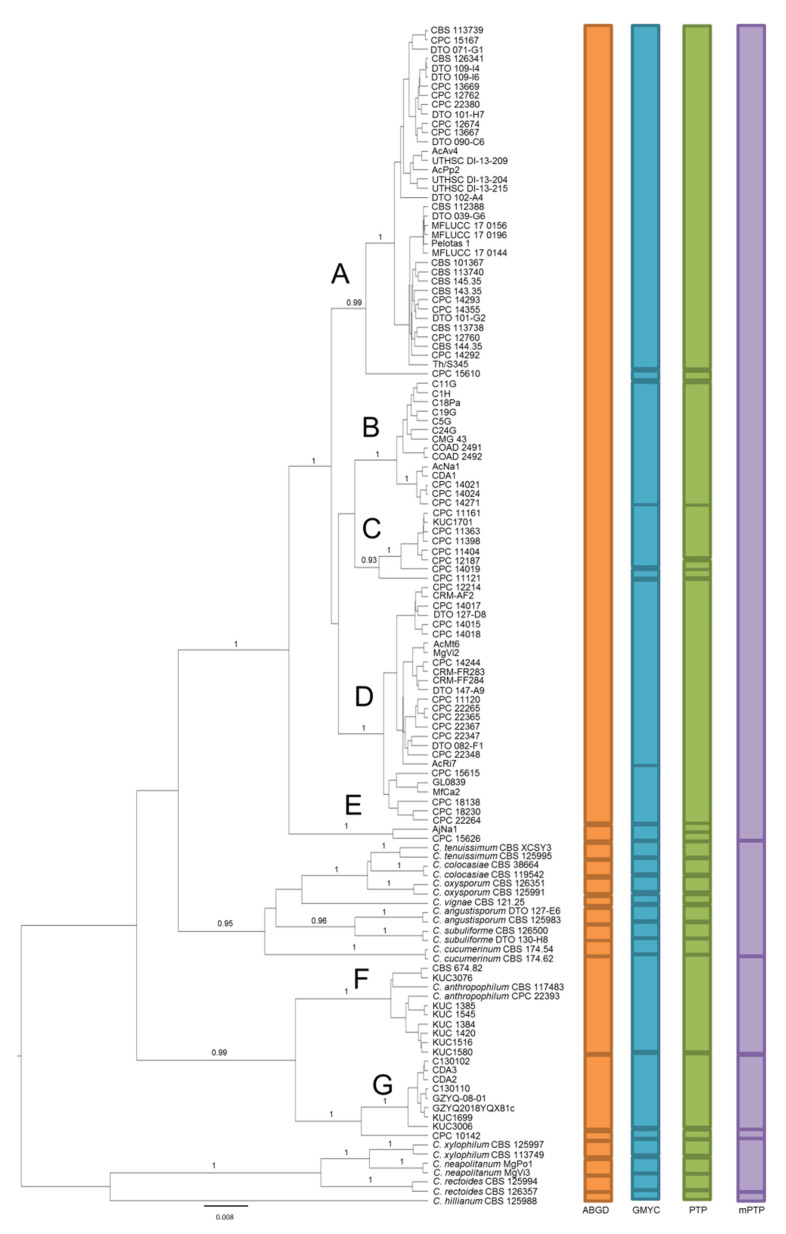
Ultrametric tree phylogeny of *C. cladosporioides* showing the results of the sequence-based species delimitation methods. The tree is the result of a Bayesian analysis performed in BEAST on the concatenated ITS, *tef1*, *act* dataset. For each node, posterior probabilities (if >0.90) are presented above the branch leading to that node. Results of species delimitation analyses are represented by colored boxes to the right. Main groups identified by phylogenetic reconstruction are indicated by letters A, B, C, D, E, F and G. The scale bar represents the substitutions per site according to the model of sequence evolution applied. Different colors indicate the different methods used.

**Table 1 pathogens-10-01167-t001:** A list of *Cladosporium cladosporioides* isolates that have a complete set of DNA barcode sequences deposited in GenBank.

Code	Source	Country	ITS	*tef1*	*act*
AcAv4	larva of *Asphondylia nepetae*	Italy	MK387888	MK416092	MK416049
AcMt6	larva of *Asphondylia nepetae*	Italy	MK387883	MK416087	MK416044
AcNa1	gall on *Clinopodium nepeta*	Italy	MK387881	MK416085	MK416042
AcPp2	*Clinopodium nepeta,* receptacle	Italy	MK387885	MK416089	MK416046
AcRi7	*Clinopodium nepeta,* receptacle	Italy	MK387886	MK416090	MK416047
AjNa1	*Ajuga reptans*, receptacle	Italy	MK387884	MK416088	MK416045
C11G	rice leaf	Brazil	MK049921	MK073937	MK073928
C130102	*Fragaria x ananassa*	Korea	KJ558398	KJ558400	KJ558395
C130110	*Fragaria x ananassa*	Korea	KJ558397	KJ558399	KJ558394
C18Pa	rice leaf	Brazil	MK049923	MK073939	MK073930
C19G	rice leaf	Brazil	MK049924	MK073940	MK073931
C1H	rice leaf	Brazil	MK049917	MK073933	MK073924
C24G	rice leaf	Brazil	MK049925	MK073941	MK073932
C5G	rice leaf	Brazil	MK049919	MK073935	MK073926
CBS 101367	soil	Brazil	HM148002	HM148243	HM148489
CBS 112388 ^†^	indoor air	Germany	HM148003	HM148244	HM148490
CBS 113738	grape bud	USA	HM148004	HM148245	HM148491
CBS 113739	crested wheat grass	USA	HM148005	HM148246	HM148492
CBS 113740	berry	USA	HM148006	HM148247	HM148493
CBS 126341	spinach seed	USA	HM148009	HM148250	HM148496
CBS 143.35	*Pisum sativum*	South Africa	HM148011	HM148252	HM148498
CBS 144.35	*Pisum sativum*	USA	HM148012	HM148253	HM148499
CBS 145.35	*Pisum sativum*	Germany	HM148013	HM148254	HM148500
CBS 674.82	cotton seed	Israel	HM148014	HM148255	HM148501
CDA1	*Phragmidium griseum*	Korea	MG451052	MG451058	MG451055
CDA2	*Gymnosporangium haraeanum*	Korea	MG451053	MG451059	MG451056
CDA3	*Gymnosporangium haraeanum*	Korea	MG451054	MG451060	MG451057
CMG 43	*Fucus spiralis*	Portugal	MN053016	MN066642	MN066637
COAD 2491	leaf litter	Brazil	MK253342	MK293782	MK249985
COAD 2492	leaf litter	Brazil	MK253343	MK293783	MK249986
CPC 10142	*Chenopodium ficifolium*	Korea	HM148015	HM148256	HM148502
CPC 11120	*Viola mandshurica*	Korea	HM148017	HM148258	HM148504
CPC 11121	*Celosia cristata*	Korea	HM148018	HM148259	HM148505
CPC 11161	*Eucalyptus* sp.	India	HM148022	HM148263	HM148509
CPC 11363	*Valeriana fauriei*	Korea	HM148023	HM148264	HM148510
CPC 11398	rust (*Phragmidium griseum*)	Korea	HM148024	HM148265	HM148511
CPC 11404	*Rubus coreanus*	Korea	HM148025	HM148266	HM148512
CPC 12187	*Myosoton aquaticum*, leaf	Korea	HM148027	HM148268	HM148514
CPC 12214	*Morus rubra*, leaf	Germany	HM148028	HM148269	HM148515
CPC 12760	*Spinacia oleracea*, seed	USA	HM148029	HM148270	HM148516
CPC 12762	*Spinacia oleracea*, seed	USA	HM148030	HM148271	HM148517
CPC 12764	*Spinacia oleracea*, seed	USA	HM148031	HM148272	HM148518
CPC 13667	*Eucalyptus robertsonii*	Australia	HM148034	HM148275	HM148521
CPC 13669	*Eucalyptus robertsonii*	Australia	HM148035	HM148276	HM148522
CPC 14015	wheat	South Africa	HM148038	HM148279	HM148525
CPC 14017	wheat	South Africa	HM148039	HM148280	HM148526
CPC 14018	wheat	South Africa	HM148040	HM148281	HM148527
CPC 14019	wheat	South Africa	HM148041	HM148282	HM148528
CPC 14021	wheat	South Africa	HM148042	HM148283	HM148529
CPC 14024	pawpaw	South Africa	HM148043	HM148284	HM148530
CPC 14244	*Magnolia* sp.	USA	HM148044	HM148285	HM148531
CPC 14271	twig of unidentified tree	France	HM148045	HM148286	HM148532
CPC 14271	unidentified tree	France	HM148045	HM148286	HM148532
CPC 14292	soil	Denmark	HM148046	HM148287	HM148533
CPC 14293	cellulose powder	Denmark	HM148047	HM148288	HM148534
CPC 14355	mouldy pea	USA	HM148048	HM148289	HM148535
CPC 15167	mite in strawberry leaf	Slovenia	HM148052	HM148293	HM148539
CPC 15610	*Rumex* sp.	Mexico	KT600385	KT600482	KT600580
CPC 15615	wild tree	Mexico	KT600386	KT600483	KT600581
CPC 15626	wild plant	Mexico	KT600387	KT600484	KT600582
CPC 18138	pine needles	Mexico	KT600388	KT600485	KT600583
CPC 18230	bract of *Phaenocoma prolifera*	South Africa	JF499834	JF499872	JF499878
CPC 22264	indoor air sample	USA	MF472936	MF473363	MF473786
CPC 22265	indoor air sample	USA	MF472937	MF473364	MF473787
CPC 22347	indoor air sample	USA	MF472938	MF473365	MF473788
CPC 22348	indoor air sample	USA	MF472939	MF473366	MF473789
CPC 22365	indoor air sample	USA	MF472940	MF473367	MF473790
CPC 22367	indoor air	USA	MF472941	MF473368	MF473791
CPC 22380	indoor air sample	USA	MF472942	MF473369	MF473792
CRM-AF2	*Vaccinium corymbosum*, fruit	Mexico	MN857901	MN865110	MN865115
CRM-FF284	*Rubus idaeus*, fruit	Mexico	MN857899	MN865108	MN865113
CRM-FR283	*Fragaria x ananassa*, fruit	Mexico	MN857900	MN865109	MN865114
DTO 039-G6	indoor air sample	Germany	KP701868	KP701745	KP701991
DTO 071-G1	indoor air sample	Greece	KP701872	KP701749	KP701995
DTO 082-F1	indoor air sample	The Netherlands	KP701879	KP701756	KP702002
DTO 090-C6	archive	The Netherlands	KP701898	KP701775	KP702021
DTO 101-G2	table	Hungary	MF472943	MF473370	MF473793
DTO 101-H7	floor	Hungary	MF472944	MF473371	MF473794
DTO 102-A4	bathroom	Hungary	KP701905	KP701782	KP702028
DTO 109-I4	indoor environment	Denmark	KP701920	KP701797	KP702043
DTO 109-I6	indoor environment	Denmark	KP701922	KP701799	KP702045
DTO 127-D8	indoor air sample	The Netherlands	KP701933	KP701810	KP702055
DTO 147-A9	indoor environment	Hungary	KP701941	KP701818	KP702063
GL0839	apple	China	JX241647	JX241672	JX241674
GZYQ-08-01	*Camellia sinensis*, leaf	China	MK852271	MK852273	MK852272
GZYQ2018YQX81c	*Camellia sinensis*, leaf	China	MK799636	MK799638	MK799637
KUC1384	Korean pine	Korea	JN033485	JN033540	JN033512
KUC1385	Korean pine	Korea	JN033484	JN033539	JN033511
KUC1420	Japanese red pine lumber	Korea	JN033483	JN033538	JN033510
KUC1516	Korean pine lumber	Korea	JN033479	JN033534	JN033506
KUC1545	Korean pine lumber	Korea	JN033478	JN033533	JN033505
KUC1580	larch	Korea	JN033477	JN033532	JN033504
KUC1699	Japanese red pine	Korea	JN033473	JN033528	JN033500
KUC1701	Japanese red pine	Korea	JN033471	JN033526	JN033498
KUC3006	radiata pine wood	Korea	JN033465	JN033520	JN033492
KUC3076	larch wood	Korea	JN033463	JN033518	JN033490
MfCa2	gall on *Micromeria fruticulosa*	Italy	MK387882	MK416086	MK416043
MFLUCC 17 0144	*Vitis vinifera*	China	MG938710	MG938823	MG938675
MFLUCC 17 0156	*Vitis vinifera*	China	MG938711	MG938824	MG938676
MFLUCC 17 0196	*Vitis vinifera*	China	MG938712	MG938825	MG938677
MgVi2	larva of *Asphondylia* sp.	Italy	MK387887	MK416091	MK416048
Pelotas1	*Alstroemeria hybrida*	Brazil	MG775703	MG775038	MG775039
Th/S345	*Thymus vulgaris*, achene	Poland	MK387889	MK416093	MK416050
UTHSC DI-13-204	abdomen	USA	LN834358	LN834454	LN834542
UTHSC DI-13-209	pleura	USA	LN834359	LN834455	LN834543
UTHSC DI-13-215	sputum	USA	LN834360	LN834456	LN834544

^†^ Ex-type from neotype of *C. cladosporioides.*

**Table 2 pathogens-10-01167-t002:** A list of 22 isolates of *Cladosporium* species used in the phylogenetic and species delimitation analyses.

Species	Code	Source	ITS	*tef1*	*act*
*C. angustisporum*	CBS 125983	*Alloxylon wickhamii*	HM147995	HM148236	HM148482
*C. angustisporum*	DTO-127-E6	air in bakery	KP701935	KP701812	KP702057
*C. anthropophilum*	CBS 117483	-	HM148007	HM148248	HM148494
*C. anthropophilum*	CPC 22393	indoor air	MF472922	MF473349	MF473772
*C. colocasiae*	CBS 386.64	*Colocasia esculenta*	HM148067	HM148310	HM148555
*C. colocasiae*	CBS 119542	*Colocasia esculenta*	HM148066	HM148309	HM148554
*C. cucumerinum*	CBS 174.62	painted floor	HM148076	HM148320	HM148565
*C. cucumerinum*	CBS 174.54	*Cucumis sativus*	HM148075	HM148319	HM148564
*C. hillianum*	CBS 125988	leaf of *Typha orientalis*	HM148097	HM148341	HM148586
*C. neapolitanum*	MgPo1	*Micromeria graeca*-receptacle	MK387890	MK416094	MK416051
*C. neapolitanum*	MgVi3	*Micromeria graeca*-receptacle	MK387892	MK416096	MK416053
*C. oxysporum*	CBS 125991	soil	HM148118	HM148362	HM148607
*C. oxysporum*	CBS 126351	indoor air	HM148119	HM148363	HM148608
*C. rectoides*	CBS 125994	*Vitis flexuosa*	HM148193	HM148438	HM148683
*C. rectoides*	CBS 126357	*Plectranthus* sp.	MH863933	HM148439	HM148684
*C. subuliforme*	CBS 126500	*Chamaedorea metallica*	HM148196	HM148441	HM148686
*C. subuliforme*	DTO-130-H8	indoor environment	KP701938	KP701815	KP702060
*C. tenuissimum*	XCSY3	*Coriandrum sativum*	MG873079	MT154184	MT154174
*C. tenuissimum*	CBS 125995	*Lagerstroemia* sp.	HM148197	HM148442	HM148687
*C. vignae*	CBS 121.25	*Vigna unguiculata*	HM148227	HM148473	HM148718
*C. xylophilum*	CBS 125997	dead wood of *Picea abies*	HM148230	HM148476	HM148721
*C. xylophilum*	CBS 113749	*Prunus avium*	HM148228	HM148474	HM148719

**Table 3 pathogens-10-01167-t003:** Taxonomic index of congruence (C_tax_) for every species delimitation method.

Method	*C* _tax_				Mean *C*_tax_
	ABGD	GMYC	PTP	mPTP	
ABGD	-	-	-	-	0.60
GMYC	0.71	-	-	-	0.62
PTP	0.68	0.87	-	-	0.60
mPTP	0.40	0.28	0.27	-	0.32

*C*_tax_ index is a measure of congruence in species assignments among two methods, with a value of 1 indicating complete congruence.

## Data Availability

The data presented in this study are openly available in Zenodo at doi: 10.5281/zenodo.5152222.

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
