# Peer review of "Cryptic Diversity in Cladosporium cladosporioides Resulting from Sequence-Based Species Delimitation Analyses"

_pathogens, 2021, doi:10.3390/pathogens10091167_

Round 1

Reviewer 1 Report

Title: perhaps add “sequenced-based species delimitation analyses” (plural analyses)

Lines 11-12: Establishing a stable taxonomy is particularly important for understanding fungal biodiversity and the evolution of particular traits related to symbiotic interactions. {replace “, as well as” by “and”}

Line 14: delete the word “genetic”

Line 16: “… the taxonomy of Cladosporium species is subjected to frequent revisions, which nowadays mostly rely on molecular data,”

The structure needs to be changed – right now this reads that the revisions rely on molecular data, not the taxonomy.

Second, taxonomy does not rely mostly on molecular data; more than ever an integrative taxonomy approach is recommended (see, e.g., Aime et al. 2021, Maharachchikumbura et al. 2021, and many others cited hererin)

Aime, M.C., Miller, A.N., Aoki, T., Bensch, K., Cai, L., Crous, P.W., Hawksworth, D.L., Hyde, K.D., Kirk, P.M., Lücking, R. and May, T.W., 2021. How to publish a new fungal species, or name, version 3.0. IMA fungus, 12(1), pp.1-15.

Maharachchikumbura, S.S., Chen, Y., Ariyawansa, H.A., Hyde, K.D., Haelewaters, D., Perera, R.H., Samarakoon, M.C., Wanasinghe, D.N., Bustamante, D.E., Liu, J.K. and Lawrence, D.P., 2021. Integrative approaches for species delimitation in Ascomycota. Fungal Diversity, pp.1-25.

Please rework this, not only in the abstract but throughout the text.

The ITS fungal barcode on its own does not provide resolution within Cladosporium species complexes (Bensch et al. 2010, Species and ecological diversity within the Cladosporium cladosporioides complex (Davidiellaceae, Capnodiales)), so why was it used in combination with the two others? While it is the fungal barcode, for Cladosporium species complexes it is not useful for species delimitation. Please rephrase.

In the introduction, please add how diverse the genus is (how many species), what their different ecological roles are, and what some of the general taxonomic issues are. I am missing a broad intro to the genus; there are five lines, and then you start talking about your own work and the aims of the paper (lines 53-70 are the aims, which is unnecessarily very long, whereas the introduction is insufficient).

Bensch, K.; Groenewald, J.Z.; Dijksterhuis, J.; Starink-Willemse, M.; Andersen, B.; Summerell, B.A.; Shin, H.D.; Dugan, F.M.; Schroers, H.J.; Braun, U.; et al. Species and ecological diversity within the Cladosporium cladosporioides complex (Davidiellaceae, Capnodiales). Stud. Mycol. 2010, 67, 1–94.

Wijayawardene, N.N., et al. 2020. Outline of Fungi and fungus-like taxa. Mycosphere, 11(1), pp.1060-1456.

Haelewaters, D., Urbina, H., Brown, S., Newerth-Henson, S. and Aime, M.C., 2021. Isolation and Molecular Characterization of the Romaine Lettuce Phylloplane Mycobiome. Journal of Fungi, 7(4), 277.

Please restructure the entire introduction. There is no logical flow. Please start broad: What is Cladosporium? Paragraph about diversity within the genus, a paragraph about ecology, and 1-2 paragraphs about taxonomy challenges. Then you could introduce species delimitation methods and how they help with taxonomy issues AS PART OF an integrative approach (refs above). And then, briefly, the goals/aims of the paper (no need for details, details need to be in the methods).

2.2 Species delimitation “analyses” -> plural

Lines 152-155: This statement appears to be a self-fulfilling prophecy: only two isolates of each species were included in the analyses. I do not know if I stand behind this statement. Please consider removing or rephrasing.

I would like to suggest to run the PTP analysis again, but only on single-locus trees. Earlier reports of species delimitation analyses say that PTP performs better when analyzing single-gene trees.

I am also wondering if it would be good to exclude ITS from the analyses. Since the ITS provides insufficient resolution at species level within these species complexes, it may also decrease the overall resolution of the concatenated tree, compared to a concatenated TEF+ACT tree. These additional analyses are worth performing.

Author Response

REVIEWER 1

Title: perhaps add “sequenced-based species delimitation analyses” (plural analyses)

We modified the title as suggested, thank you!

Lines 11-12: Establishing a stable taxonomy is particularly important for understanding fungal biodiversity and the evolution of particular traits related to symbiotic interactions. {replace “, as well as” by “and”}

Done.

Line 14: delete the word “genetic”

Done.

Line 16: “… the taxonomy of Cladosporium species is subjected to frequent revisions, which nowadays mostly rely on molecular data,”

The structure needs to be changed – right now this reads that the revisions rely on molecular data, not the taxonomy.

Second, taxonomy does not rely mostly on molecular data; more than ever an integrative taxonomy approach is recommended (see, e.g., Aime et al. 2021, Maharachchikumbura et al. 2021, and many others cited hererin)

Aime, M.C., Miller, A.N., Aoki, T., Bensch, K., Cai, L., Crous, P.W., Hawksworth, D.L., Hyde, K.D., Kirk, P.M., Lücking, R. and May, T.W., 2021. How to publish a new fungal species, or name, version 3.0. IMA fungus, 12(1), pp.1-15.

Maharachchikumbura, S.S., Chen, Y., Ariyawansa, H.A., Hyde, K.D., Haelewaters, D., Perera, R.H., Samarakoon, M.C., Wanasinghe, D.N., Bustamante, D.E., Liu, J.K. and Lawrence, D.P., 2021. Integrative approaches for species delimitation in Ascomycota. Fungal Diversity, pp.1-25.

Please rework this, not only in the abstract but throughout the text.

Thanks for your suggestion! We changed the abstract as follows: “Although the taxonomy of Cladosporium species, which nowadays mostly rely on the integration of molecular data with morphological and cultural characters, is subjected to frequent revisions the recently developed species delimitation methods have not been used to explore cryptic diversity in this genus.”

Moreover we edited this concept throughout the text and added the suggested references.

The ITS fungal barcode on its own does not provide resolution within Cladosporium species complexes (Bensch et al. 2010, Species and ecological diversity within the Cladosporium cladosporioides complex (Davidiellaceae, Capnodiales)), so why was it used in combination with the two others? While it is the fungal barcode, for Cladosporium species complexes it is not useful for species delimitation. Please rephrase.

Although ITS does not provide a good species resolution, it is still informative and historically combined with TEF1 and ACT markers in milestone and recent papers about Cladosporium genus (Bensch et al. 2012, Bensch et al. 2015, Iturrieta et al 2021). In order to facilitate the comparison with the phylogenies shown in these papers and our previous article (Zimowska et al 2021), we preferred to include the ITS data in the present work.

In the introduction, please add how diverse the genus is (how many species), what their different ecological roles are, and what some of the general taxonomic issues are. I am missing a broad intro to the genus; there are five lines, and then you start talking about your own work and the aims of the paper (lines 53-70 are the aims, which is unnecessarily very long, whereas the introduction is insufficient).

Bensch, K.; Groenewald, J.Z.; Dijksterhuis, J.; Starink-Willemse, M.; Andersen, B.; Summerell, B.A.; Shin, H.D.; Dugan, F.M.; Schroers, H.J.; Braun, U.; et al. Species and ecological diversity within the Cladosporium cladosporioides complex (Davidiellaceae, Capnodiales). Stud. Mycol. 2010, 67, 1–94.

Wijayawardene, N.N., et al. 2020. Outline of Fungi and fungus-like taxa. Mycosphere, 11(1), pp.1060-1456.

Haelewaters, D., Urbina, H., Brown, S., Newerth-Henson, S. and Aime, M.C., 2021. Isolation and Molecular Characterization of the Romaine Lettuce Phylloplane Mycobiome. Journal of Fungi, 7(4), 277.

Please restructure the entire introduction. There is no logical flow. Please start broad: What is Cladosporium? Paragraph about diversity within the genus, a paragraph about ecology, and 1-2 paragraphs about taxonomy challenges. Then you could introduce species delimitation methods and how they help with taxonomy issues AS PART OF an integrative approach (refs above). And then, briefly, the goals/aims of the paper (no need for details, details need to be in the methods).

Thanks for your important suggestions, the Introduction has been extensively rephrased and integrated with the missing details on ecology and phylogeny.

2.2 Species delimitation “analyses” -> plural

Edited in the text.

Lines 152-155: This statement appears to be a self-fulfilling prophecy: only two isolates of each species were included in the analyses. I do not know if I stand behind this statement. Please consider removing or rephrasing.

Lines 152-155 state: ‘The tested species delimitation methods correctly identified known species, except for mPTP, which failed in discriminating among C. tenuissimum, C. colocasiae, C. oxysporum, C. vignae, C. angustisporum, C. subuliforme and C. cucumerinum, as well as among C. xylophilum, C. neapolitanum and C. rectoides. This is a part of results regarding the sensitivity of the methods used in our work. We sincerely do not understand where is the “self-fulfilling prophecy” here.

I would like to suggest to run the PTP analysis again, but only on single-locus trees. Earlier reports of species delimitation analyses say that PTP performs better when analyzing single-gene trees.

I am also wondering if it would be good to exclude ITS from the analyses. Since the ITS provides insufficient resolution at species level within these species complexes, it may also decrease the overall resolution of the concatenated tree, compared to a concatenated TEF+ACT tree. These additional analyses are worth performing.

We excluded this kind of analysis because the official methods for Cladosporium are based on concatenated sequences. Thus we preferred considering the same dataset of combined loci for all our analyses in order to be consistent, and to allow a direct comparison with previous studies.

Reviewer 2 Report

Dear Authors,

in my opinion your work is very interesting in a cognitive context and contributes a lot to mycology, molecular and evolutionary taxonomy.

All the tables and figures are appropriate for this type of article. In general, the paper has a logical flow and it is refined in detail. The abstract well correspond with the main aspects of the work. Nevertheless, I see a few and non-significant weak points in this work (given below), which I am convinced that the Authors are able to resolve very fast.

As a reviewer I am obligated to pay attention even to less important weak points of this work and all mentioned below comments should be carefully considered.

The Authors use the abbreviation ACT throughout the manuscript to refer to actin gene. It would be good, however, to specify whether all information contained in the paper relates specifically to ACT1 gene. In addition, abbreviations (gene names) should be written in italics.

Line 12

In my opinion sounds better if instead of ,, ... as well as the evolution of particular traits related to symbiotic interactions. " Authors will use: ,, as well as the evolution of particular traits related to interspecific interactions." which will be more adequate with the actual state of affairs. In the next sentence, the Authors write about "associations ranging from mutualistic to pathogenic" and not only about "symbiotic interactions"

Line 14

In my opinion ,,…most frequently represented…” sounds better

Line 34

Should be ,,biocenotic”

Line 40

There is ,,taxonomic grouping” but should be ,,taxonomic groups”

Line 52

There is ,,molecular tool” but in my opinion should be ,,molecular tools”

Lines 57-58

In my opinion the sentence ,,required for molecular delimitation at the species level by multilocus approach” sounds better

Lines 152-153

Can the information, relating to the results, posted in the sentence (quote) "The tested species delimitation methods correctly identified known species, except for mPTP, which failed in discriminating among ..." be unequivocally explained by the fact that (as the Authors themselves stated) ,,mPTP is the most conservative method "or maybe explanation of this is more complicated. In my opinion it would be of benefit to potential readers to refer to it in the Discussion.

Lines 173-174

My suggestion is to change this part of sentence ,,… we explored cryptic diversity in the group of all the considered C. cladosporioides strains”

Line 206

,,mean Ctax index values” sounds better

Line 222

Should be ,,indicate one species”

Line 238 and 244

,,which is ascribed as C. antropophilum” sounds better

Line 252

I think that in this place it is worth briefly justifying why Muscle program was used instead of CLUSTALW for alignment of sequences

Line 174 and 275

How the information about sequence data for ITS, TEF1 and ACT, which are available in GenBank (last accessed in May 2021) relates to the statement: ,,Phylogenetic trees were drawn by using FigTree software (tree.bio.ed.ac.uk/software / figtree /) (accessed on 10 December 2020).” Is this information compatible despite the time discrepancy?

Line 316

Correct is ,,kingdom of Fungi”, not ,,realm of Fungi”

Author Response

REVIEWER 2

The Authors use the abbreviation ACT throughout the manuscript to refer to actin gene. It would be good, however, to specify whether all information contained in the paper relates specifically to ACT1 gene. In addition, abbreviations (gene names) should be written in italics.

Thanks for your suggestions, we added the information on act1 in the abstract and in the introduction, specifying that we refer to act1 gene throughout the manuscript. Moreover we used italics for referring to the genes.

Line 12

In my opinion sounds better if instead of ,, ... as well as the evolution of particular traits related to symbiotic interactions. " Authors will use: ,, as well as the evolution of particular traits related to interspecific interactions." which will be more adequate with the actual state of affairs. In the next sentence, the Authors write about "associations ranging from mutualistic to pathogenic" and not only about "symbiotic interactions"

We edited the phrase using the word “interspecific” as suggested.

Line 14

In my opinion ,,…most frequently represented…” sounds better

Done.

Line 34

Should be ,,biocenotic”

Done.

Line 40

There is ,,taxonomic grouping” but should be ,,taxonomic groups”

Done.

Line 52

There is ,,molecular tool” but in my opinion should be ,,molecular tools”

Done.

Lines 57-58

In my opinion the sentence ,,required for molecular delimitation at the species level by multilocus approach” sounds better

Done.

Lines 152-153

Can the information, relating to the results, posted in the sentence (quote) "The tested species delimitation methods correctly identified known species, except for mPTP, which failed in discriminating among ..." be unequivocally explained by the fact that (as the Authors themselves stated) ,,mPTP is the most conservative method "or maybe explanation of this is more complicated. In my opinion it would be of benefit to potential readers to refer to it in the Discussion.

Thanks. We edited the Discussion section as suggested (lines 204-208): “Surprisingly, the multi-rate extension of PTP (mPTP) cannot discriminate among known species revealing an exaggerated lumping tendency in our conditions. Indeed, this method greatly differs from the others in terms of mean values of Ctax index, which is a measure of reciprocal congruence between methods, revealing poor performance in species delimitation for the considered species. These findings are in line with a previous work which suggested that mPTP method is more conservative than GMYC [25].”

We feel that a more in depth hypothesis on the origin of such failure of mPTP in distinguishing among species requires further studies.

Lines 173-174

My suggestion is to change this part of sentence ,,… we explored cryptic diversity in the group of all the considered C. cladosporioides strains”

Done.

Line 206

,,mean Ctax index values” sounds better

Done.

Line 222

Should be ,,indicate one species”

Done.

Line 238 and 244

,,which is ascribed as C. antropophilum” sounds better

We think “ascribed to C. anthropophilum” is correct.

Line 252

I think that in this place it is worth briefly justifying why Muscle program was used instead of CLUSTALW for alignment of sequences

ClustalW produces more gaps when nucleic acid sequences lengths are quite different (as in our case) so we preferred MUSCLE, which is claimed to achieve both better average accuracy and better speed than ClustalW or T-Coffee.

REFERENCES:

  • https://www.ebi.ac.uk/Tools/msa/muscle/
  • Edgar, R.C. MUSCLE: a multiple sequence alignment method with reduced time and space complexity. BMC Bioinformatics 5, 113 (2004). https://doi.org/10.1186/1471-2105-5-113

However we do not feel that we need to justify this choice in the methods.

Line 174 and 275

How the information about sequence data for ITS, TEF1 and ACT, which are available in GenBank (last accessed in May 2021) relates to the statement: ,,Phylogenetic trees were drawn by using FigTree software (tree.bio.ed.ac.uk/software / figtree /) (accessed on 10 December 2020).” Is this information compatible despite the time discrepancy?

Yes, FigTree is just a software that we downloaded on December 2020 and we used to plot the trees obtained with the molecular data obtained on May 2021. However, in order to avoid any misunderstanding we decided to remove the part “(accessed on 10 December 2020)”, because it refers to a software and not to a database.

Line 316

Correct is ,,kingdom of Fungi”, not ,,realm of Fungi”

We know that Kingdom is the correct word but we preferred using “realm” which is referred to Bensch et al 2015 article titled: “Common but different: The expanding realm of Cladosporium” and represents a more creative way to connect our work with previous findings.

Reviewer 3 Report

Review on pathogens-1346464
manuscript entitled “Cryptic Diversity in Cladosporium cladosporioides Resulting from
Species Delimitation Analysis” by Andrea Becchimanzi, Beata Zimowska and Rosario Nicoletti

This study is in general a valuable addendum to the previously published work by Zimowska et al. (2021, also in Pathogens), dealing with Cladosporium cladosporioides spp. complex.

It is really surprising that this species complex still harbors so many good phylogenic species waiting for formal description. The authors intention here was not to describe new species rather to point at the genetic diversity within the C. cladosporioides complex, and give a clear phylogenetic evidence of an existence of putatively novel taxa. I think it is all fine, although I miss a bit a true mycology here (in sense of description of new taxa, phenotypic comparison, differential species diagnoses, pictures and micro-photos…). The genus Cladosporium is indeed an incredibly diverse "realm" worth to explore further, not only from taxonomical point of view. The phylogenetic analyses used in the current work sound really highly convincing, providing solid data to the final taxonomic concept into the genus. This is some good piece of work indeed.

As for the detection of the presence of a 60 pb intron in TEF sequences, this is very interesting approach. However, it is not clear to me how these parts were detected (please clarify into Material and Method).

To the manuscript some additional formal comments:

  1. Table 1and2 Captions/Figure 1 should contain all abbreviations of fungal collections with their full names for all strains listed;
  2. ex-neotype and/or ex-type cultures should be indicated;
  3. indicate also the outgroup used into phylotree and it should not be a taxon of C. cladosporioides complex (so no C. hillianum).
  4. Indicate please also ex-type cultures in phylo trees (and ex-neotype of C. cladosporioides),
  5. Use another rooting species (e.g. from C. herbarum complex).
  6. The authors claim about the erroneous taxonomical assignment of several strains (group F) that should be considered as recently described C. anthropopillum (e.g. strains KUC 1545, KUC 1580, CBS 674.82 etc). Could you please provide %similarity ITS, TEF and ACT loci to the ex-type culture of C. anthropophilum at least for some of those "erroneously assigned" strains.
  7. Why the phylotree does not contain C. polonicum and C. pseudocladosporioides? This is optional. Would be fine to see where these two taxa are resolved and to promote the authors newly described taxon from Poland too. (optional, of course)
  8. Line 127 … ..our previous work, and …. should be …. our previous work [3], and..
  9. Line 179 …. to tester strains of … should be … to reference strains of …

----- end----

Author Response

REVIEWER 3

As for the detection of the presence of a 60 pb intron in TEF sequences, this is very interesting approach. However, it is not clear to me how these parts were detected (please clarify into Material and Method).

The following sentence was added to Materials and Methods section: “The aligned sequences were manually checked in order to identify introns, which are frequent in tef1 [17] and are characterized by the presence of GT-AG nucleotides (5’-3’).”

To the manuscript some additional formal comments:

  1. Table 1and 2 Captions/Figure 1 should contain all abbreviations of fungal collections with their full names for all strains listed;

Done.

  1. ex-neotype and/or ex-type cultures should be indicated;

Done.

  1. indicate also the outgroup used into phylotree and it should not be a taxon of cladosporioides complex (so no C. hillianum).

We used C. hillianum to allow a better comparison with our previous work (Zimowska et al., 2021), which showed that this species is phylogenetically far enough to be used as an outgroup, leading to correct known species identification by species delimitation software.

  1. Indicate please also ex-type cultures in phylo trees (and ex-neotype of cladosporioides),

Done.

  1. Use another rooting species (e.g. from herbarum complex).

See our response to comment 3.

  1. The authors claim about the erroneous taxonomical assignment of several strains (group F) that should be considered as recently described anthropopillum (e.g. strains KUC 1545, KUC 1580, CBS 674.82 etc). Could you please provide %similarity ITS, TEF and ACT loci to the ex-type culture of C. anthropophilum at least for some of those "erroneously assigned" strains.

We performed such Blast analyses as suggested and reported here the results:

Loci identity via Blastn

CBS 674.82 vs CBS 117483 (C. anthropophilum)

ITS =  100%

TEF= 98.46%

ACT= 98.2%

KUC 1545 vs CBS 117483 (C. anthropophilum)

ITS =  99.79%

TEF= 96.57%

ACT= 99.03%

KUC 1545 vs CPC 22393 (C. anthropophilum)

ITS = 100%

TEF = 97.03%

ACT = 100%

KUC 1580 vs CPC 22393 (C. anthropophilum)

ITS = 99.8%

TEF = 99.5%

ACT = 99.03%

As expected, reciprocal identity of single loci among C. anthropophilum and C. cladosporioides isolates in the same group (group F) is pretty high. These data are deducible from the phylogenetic tree so we do not feel that is useful to add them to the Results section.

  1. Why the phylotree does not contain polonicum and C. pseudocladosporioides? This is optional. Would be fine to see where these two taxa are resolved and to promote the authors newly described taxon from Poland too. (optional, of course)

After an all-encompassing analysis with all the C. cladosporioides species complex (data not shown), we focused on species more closely related to C. cladosporioides, and excluded relatively far species, which were not informative, from the analyses (see Zimowska et al., 2021).

  1. Line 127 … ..our previous work, and …. should be …. our previous work [3], and..

Done.

  1. Line 179 …. to tester strains of … should be … to reference strains of …

Done.

Round 2

Reviewer 1 Report

Dear authors, 

Please have the manuscript checked by an English-native speaker. I think the revised manuscript is improved. It still has a number of weaknesses, including the introduction that is still not very good, but my suggestions to improve were in the previouw review round. I do think the manuscript should be checked by an English native person; some of the language is clunky/awkwardly phrased. 

Author Response

Now I submit the new version of our manuscript after extensive revision concerning English language.
Than you and best regards,
Beata Zimowska
